# A Controlled Trial in Community Pediatrics to Empower Parents Who Are at Risk for Parenting Stress: The Supportive Parenting Intervention

**DOI:** 10.3390/ijerph16224508

**Published:** 2019-11-15

**Authors:** Amy van Grieken, Esther M.B. Horrevorts, Cathelijne L. Mieloo, Rienke Bannink, Merian B.R. Bouwmeester-Landweer, Esther Hafkamp-de Groen, Suzanne Broeren, Hein Raat

**Affiliations:** 1Department of Public Health, Erasmus University Medical Center, 3000 CA Rotterdam, The Netherlands; 2Municipality of Rotterdam, Research and Business Intelligence, 3072 AP Rotterdam, The Netherlands; 3Center for Youth and Family, 3003 AB Rotterdam, The Netherlands; 4Netherlands Centre for Youth Health Care, 3527 GV Utrecht, The Netherlands; 5Rivas Zorggroep, 4205 JC Gorinchem, The Netherlands

**Keywords:** parenting stress, child psycho-social development, intervention, pediatrics

## Abstract

The goal of the Supportive Parenting Intervention is to prevent and/or decrease parenting stress and provide a sense of empowerment to parents with a newborn child. We evaluated the effectiveness of the Supportive Parenting Intervention in terms of parenting skills, social support, self-sufficiency, resilience, and child psychosocial health. A controlled trial with pre- and post-intervention testing was conducted in the setting of community pediatrics among parents at risk for developing parenting stress. The 177 parents in the control group received care-as-usual, whereas the 124 parents in the intervention group received six home visits by a trained Youth Health Care nurse during the first 18 months of the child’s life. The result with respect to parenting skills, social support (both from family and friends, and the partner), self-sufficiency, and resilience at the 18-month follow-up was either unchanged or (*p* < 0.05) worse compared to the respective baseline score for both groups. We found no significant difference between intervention and control group with respect to the child’s Child Behavior Check List (CBCL). This study shows no positive effect with respect to the indicators of parental empowerment. We recommend research to strengthen the intervention and its application in daily practice, for example by increasing the intervention duration, and to evaluate it in a large randomized controlled trial.

## 1. Background

Becoming a first-time parent can be an overwhelming experience, providing many unanticipated outcomes. Research has shown that many parents and caregivers (collectively referred to here as “parents”) have questions and self-doubt, particularly when the child is young (e.g., in the infant and toddler stages). Concerns regarding parenting in general and the child’s developmental progress and/or behavior are common among parents of children around 14 months of age [1]. Indeed, nearly 60% of these parents reported having parental concerns for which they felt they needed assistance or advice from someone outside of the family and approximately 11% of parents report having frequent concerns [1]. The circumstances in the parents’ lives can cause parenting to become even more challenging and stressful [2]. Östberg conceptualized parenting stress as the perceived discrepancy between the situation’s requirements and the parents’ personal resources [3]. Parenting stress can lead to severe parenting practices such as child abuse and neglect [4], both of which can lead to adverse physical, cognitive, and psychosocial outcomes for the child in both the short and long terms [5,6,7]. Circumstances that can contribute to parenting stress include a lack of social support [8,9], being a single parent [10], being a young parent (<19 years) [8,9,10,11], ambivalent feelings towards becoming a parent [9,12], depressive feelings [8,9,11], spousal violence [8,9,13,14,15], alcohol and/or drug abuse [8,9,16], negative experiences during childhood [8,9,11,12,13,17,18], negative sexual experiences [11,12,18], the tendency to become upset and/or angry [8], low birth weight of the child [19], and preterm birth [19]. 

In the Netherlands, community-based pediatric care is well organized and focuses on monitoring the child’s growth, health, well-being, behavior, and development by promoting health, and preventing disease; this preventive service is known collectively as Youth Health Care (YHC). Until the age of 19 years, each child in the Netherlands is invited by the YHC to undergo periodic health examinations, which usually take place at a YHC center or at the child’s school; these health examinations are provided free of charge [20]. Although participation is voluntary, the compliance rate within the first months of childbirth is approximately 95% [20,21]. Importantly, the YHC’s goal is to support parents in the development of their parenting skills and to promote the healthy development of all children. Therefore, the YHC contributes to the prevention and to the early detection of parents who experience parenting stress; moreover, the YHC offers suitable interventions to these parents. 

One such preventive intervention available to YHC centers is the Supportive Parenting Intervention. This intervention is designed to decrease parenting stress and empower parents with a newborn child who are at risk for developing parenting stress [22]. The risk assessment tool IPARAN (Instrument for identification of Parents At Risk for child Abuse and Neglect) is used to identify at-risk parents based on the parents’ previous negative experiences (for example, during their own childhood) and current circumstances such as ambivalent feelings regarding becoming a parent and a lack of social support. A trained YHC nurse uses the IPARAN to identify parents who are at risk for developing parenting stress and are therefore eligible to receive the Supportive Parenting Intervention. This intervention consists of six 90-minute home visits by a YHC nurse during the first 18 months of the child’s life. Currently, approximately half of all YHC centers in the Netherlands offer the Supportive Parenting Intervention to parents.

A previous study [22] assessed the effectiveness of the Supportive Parenting Intervention with respect to the child’s psychosocial development, the parents’ expectations and empathic awareness of their children’s needs, the potential for child abuse, and family support. The authors found significant positive effects in terms of the parents’ expectations, the child’s psychosocial development, and the potential for child abuse [22].

In this paper we present the results of a more elaborate controlled trial designed to assess effects of the program on the factors targeted by the intervention. Parents who are at risk for developing parenting stress were compared to parents who had the same risk but received care as usual. We evaluated the intervention’s effects on parenting skills, social support (both from family and friends, and from the partner), self-sufficiency, and resilience. In addition, we examined the intervention’s effect on the child’s health, specifically child psychosocial health measured by the Child Behavior Check List (CBCL). We investigated which characteristics of the parents and child affected the Supportive Parenting Intervention’s ability to empower at-risk parents. Finally, we evaluated the experiences of the parents who received the Supportive Parenting Intervention.

## 2. Methods

### 2.1. Study Design and Power of the Study

In this study, a non-randomized controlled trial was performed with an intervention group and a control group. The design of this study has been described in detail elsewhere [23]. The trial was registered with the Netherlands Trial Register retrospectively (www.trialregister.nl; NTR 5307). This study was approved by the Medical Ethics Committee of Erasmus Medical Center Rotterdam (MEC-2013-568). Because the Supportive Parenting Intervention is currently not available at all YHC centers in the Netherlands, the intervention group was recruited in an area in which the intervention was available. The control group was recruited in an area in which the Supportive Parenting Intervention has not been available since January 2014; the control group received care as usual. 

An opportunity sample of two preventive YHC centers (Centre for Youth and Family (CJG) Rijnmond and Rivas Zorggroep) in two regions of the Netherlands participated in this study. All the parents who started the Supportive Parenting Intervention between January 2014 and September 2014 (*n* = 392) were invited by CJG Rijnmond to participate in the intervention group. All the parents with a child born between January 2014 and September 2014 who were living in an area in which the Supportive Parenting Intervention was not available (*n* = 2659) were invited by Rivas Zorggroep to participate in the control group. Taking into account informed consent by 50% and eligibility of 10% of the parents in the control group, we expected data of 196 parents in the intervention group and 133 parents in the control group, a total of 329 parents. This was slightly more, as described in the design paper [23].

With the use of continuous measures and assuming a standard deviation of 1.00 in both study groups, a power of 0.80 and an alpha of 0.05, and being able to demonstrate a significant difference of 0.35 standard deviation units of continuous outcome measures between the intervention and control group, a total group size of 329 was appropriate to indicate relevant effects [24,25].

### 2.2. Participants and Data Collection

For the intervention group, all the parents of a newborn child received, within 5–14 days of childbirth, the Instrument for identification of Parents At Risk for child Abuse and Neglect (IPARAN) in order to identify risk factors associated with the development of parenting stress (a detailed description of the IPARAN is provided in the Appendix A). Based on their IPARAN score, at-risk parents were offered the Supportive Parenting Intervention. Parents were free to refuse the intervention. In total, 392 parents who started the Supportive Parenting Intervention between January and September 2014 were invited to participate in the study and were given an information leaflet; 124 parents provided written informed consent and completed the baseline questionnaire (a response rate of 31.6%).

For the control group, a total of 2659 parents were invited to participate and were given an information leaflet; 759 parents provided written informed consent and completed the baseline questionnaire (a response rate of 28.5%). Of these 759 parents, 177 (23.3%) were eligible to participate in the control group, based on their IPARAN score.

Data were collected at baseline, when the child was 12 months of age, and at the end of the intervention (when the child was 18 months of age). At baseline, the five primary outcomes of parental empowerment (parenting skills, social support (both from family and friends, and from the partner), self-sufficiency, and resilience) were measured. At the 18-month time point, the five primary outcomes of parental empowerment were measured again, as was the child’s psychosocial health; in addition, the parents in the intervention group were also requested to answer questions regarding whether they appreciated the Supportive Parenting Intervention. 

Parents completed the baseline and follow-up questionnaire in their own time, the nurse was not present. At the 18-month time point, the questionnaire was completed by 85/124 (69%) of the parents in the intervention group and by 149/177 (84%) of the parents in the control group. In this study, we evaluated the effect of the Supportive Parenting Intervention on the parental empowerment by comparing data of the baseline with data at 18 months follow-up. 

The enrollment, allocation, and participation of parents in the study are summarized in Figure 1.

### 2.3. Supportive Parenting Intervention

The Supportive Parenting Intervention is based on the ecological model of Belsky [26,27,28] and Newberger’s concept of parental awareness [29], which was elaborated upon by Baartman [30]. More detail on the development of the intervention can be found in the thesis of Bouwmeester-Landweer [22]. The intervention is designed to decrease stress among parents with a newborn child, specifically among parents who are at risk for developing parenting stress based on their IPARAN scores. Parents are considered to be at risk when they present with one or more of the following risk factors: lack of social support; being a single parent; being a relatively young parent; negative childhood experiences; feelings of depression; alcohol and/or drug abuse; ambivalent feelings towards parenthood; the tendency to become easily upset and/or angry; negative sexual experience; in favor of physical punishment; having a child born pre-term and/or with low birth weight. 

The intervention offered to at-risk parents is voluntary and consists of six 90-minute home visits by a YHC nurse during the first 18 months of the child’s life. Although the standard schedule calls for the six home visits to occur at 6 weeks and 3, 6, 9, 12, and 18 months of age, at the parents’ request, more visits can occur during the first several months, with fewer visits toward the end of the intervention. A home visit consists of a fixed part and a flexible part. During the fixed part of the visit, various topics are discussed, including empowering the parents by improving their parenting skills, discussing the parents’ expectations regarding being a parent, and providing advice to the parents regarding how to improve their social network. The flexible part of the visit is client-centered, and the topics discussed are chosen by the parents and can include empowerment issues and troubling experiences regarding parenting. In general, the focus of these home visits is to improve the interaction between the parent and the child, discussing the parents’ expectations regarding parenthood, dealing with the parents’ own childhood experiences, and building social support around the family. The YHC nurses who conduct the home visits have at least three years of experience in the YHC system and have received additional training in order to provide the Supportive Parenting Intervention. Thus, these nurses have the knowledge needed in order to provide parents with information regarding both health- and development-related issues. However, the nurses are not qualified to provide psychotherapy treatment or family therapy; if deemed necessary, the YHC nurse can refer the parents to a specialist for more extensive treatment [31].

### 2.4. Care as Usual

During the study, the parents in the control group lived in a region in which the Supportive Parenting Intervention was not offered by their YHC center. These parents were therefore offered care as usual, in which the parents were invited to bring their child to the YHC for periodic health examinations. Twelve such examinations are offered within the first 18 months of the child’s life, during which the child’s growth and development are monitored, and the parents receive general advice regarding parenting. If deemed necessary, the parents can be referred to a specialist (e.g., to receive a home visit by a trained YHC nurse, referral to a child psychologist, or invited to attend a course in parenting).

### 2.5. Ethics Statement

All the questionnaires were completed by the parents on a voluntary basis. Parents received detailed written information regarding the study and were free to decline participation. Parents who chose to participate in the study provided written informed consent. Only anonymous data were used for analysis. The study was approved by the Medical Ethics Committee of Erasmus Medical Center, Rotterdam (MEC-2013-568).

### 2.6. Measurements

#### 2.6.1. Primary Outcome Measurements

The primary outcomes of this study were five aspects of empowerment (the parents’ parenting skills, social support (both from family and friends, and from the partner), self-sufficiency, and resilience and the child’s psychosocial health at 18 months of age. All the outcomes were collected through questionnaires.

Parenting skills was measured using the subscale “basic parenting skills” of the Family Functioning Questionnaire (FFQ) [32]. The FFQ assesses the problems that parents encounter within their family and provides an indication of how well the parent believes that his/her family functions, and consists of five subscales in total. The subscale basic parenting skills assesses the extent to which the parent feels capable of managing the household and finances, maintaining contact with the daycare provider (if applicable), and provides for the child’s daily needs. The subscale consists of seven items that are answered using a 4-point response scale ranging from 1 (not true) to 4 (very true). The individual item scores were summed to obtain a total score [32]. The subscale “basic parenting skills” had a Cronbach’s alpha value of 0.72. 

Social support from family and friends was measured using the “social contacts” subscale of the FFQ. Social support from the partner was measured using the “partner relationship” subscale of the FFQ. The “social contacts” subscale assesses the extent to which the parent has contact with his/her neighbors, friends, and family members. The subscale “partner relationship” assesses a person’s perception of his/her relationship with his/her partner and the extent to which he/she feels supported by his/her partner. Each subscale contains five items that are answered using a 4-point response scale ranging from 1 (not true) to 4 (very true). For each subscale, the individual item scores were summed to obtain a total score [32]. The subscales “social contacts” and “partner relationship” had a Cronbach’s alpha value of 0.74 and 0.89, respectively. The total scores of both subscales were used as outcome measures.

Self-sufficiency was measured using version 2.0 of the Dutch Empowerment Questionnaire for parents [33]. This questionnaire consists of three subscales: “competence as a person”, “competence as a parent”, and “competence utilization”. The subscale “competence as a parent” (7 items) was used as the primary outcome measure. Each item in the subscale is scored using a 5-point response scale ranging from 1 (strongly disagree) to 5 (strongly agree). Three items in the subscale were not included in the questionnaire, as these items were not yet applicable to parents of a newborn child. The individual item scores were summed and the total score was then converted to a score ranging from 1 to 10. A low score indicates problems [33]. The subscale “competence as a parent” had a Cronbach’s alpha value of 0.79.

The parents’ resilience was measured using the Dutch version of the Resilience Scale [34]. This questionnaire consists of a total of 25 items in the subscales “personal competence” (17 items) and “acceptance of self and life” (8 items). The subscale “personal competence” assesses the extent to which a person can rely on himself/herself, is determined to achieve a goal, and is resourceful [35]. The subscale “acceptance of self and life” assesses a person’s adaptability, flexibility, and balanced perspective on life [35]. Each item is scored on a 4-point scale ranging from 1 (strongly disagree) to 4 (strongly agree). For each subscale, the individual scores were summed, and a total score was calculated by adding the two summed scores from the two subscales. Thus, the total score ranges from 25 to 100, with a higher score indicating higher resilience [34]. This questionnaire had a Cronbach’s alpha value of 0.89.

The child’s behavioral and emotional problems were assessed at 18 months of age using the Child Behavior Checklist (CBCL) for ages 1½–5 [36]. The CBCL consists of 99 items, each of which is scored on a 3-point scale ranging from 1 (not true) to 3 (very true or often true) plus one open-ended item for adding problems that are not mentioned on the form. The CBCL has two domains (Internalizing, and Externalizing) and provides a Total Problem score. A higher score indicates problems [36]. The two domains Internalizing, and Externalizing, and the Total Problem score had a Cronbach’s alpha value of 0.85, 0.89, and 0.94, respectively.

#### 2.6.2. Parent and Child Characteristics

Data regarding the age, gender, and country of birth of the child and both parents were collected. The education level of both parents, the family structure, and the net monthly family income were also collected, as well as data regarding the gestational age, and the child’s birth weight. The parent’s level of education was classified as low (primary education or lower secondary education), middle (higher secondary education or vocational education), or high (higher vocational education or university). Net family income was classified as low (less than €1800/month) or high (€1800/month or higher). The nationality of the child and both parents was classified as either Dutch or non-Dutch using the definitions established by Statistics Netherlands [37]. 

#### 2.6.3. Other Measures

The parents in the intervention group answered questions regarding the number of home visits they received and they were asked six questions regarding what they learned from the Supportive Parenting Intervention. The questionnaire also contained an open question inviting parents to write any comments and/or questions with respect to the study and/or intervention. 

### 2.7. Statistical Analyses

Data were analyzed according to the intention-to-treat principle. Descriptive statistics were used to describe the sample characteristics. The chi-square test and independent Student’s *t*-test were used to test for differences in the demographic characteristics between the intervention and control groups. In addition, we tested for differences between participants, completing both questionnaires at baseline and follow-up at 18 months, and participants lost-to-follow up (i.e., participants only completing the baseline questionnaire). Descriptive statistics were also used to describe the parents’ evaluation of the Supportive Parenting Intervention. 

Complete cases at baseline and follow-up at 18 months were used to evaluate the within-group change between baseline and follow-up with respect to the primary outcomes. A paired Student’s *t*-test was used. To evaluate differences in primary outcomes between the intervention and control groups, difference scores (i.e., the difference in mean scores between baseline and follow-up) were calculated, and generalized linear models were used, with the study group (i.e., intervention or control group) as the independent variable and the difference scores for the primary outcomes as the dependent variables. Beta values with a 95% confidence interval (95% CI) were calculated, and beta values with a *p*-value < 0.05 were considered statistically significant. 

Potential confounders were added to the model to correct for confounding. Confounders were selected based on a significantly different distribution (categorical variables) or the average (continuous variables) in characteristics between the intervention and control groups (Table 1), and based on having a significant association with the outcome of interest. The following confounders were included in the models predicting parenting skills, social support (from family and friends, and from the partner), self-sufficiency, and resilience: baseline scores, the age of the child, the nationality of the mother and the mother’s IPARAN score. Models with correction for potential confounders and models without correction for potential confounders are presented. Based on the aforementioned evaluations and a *p*-value < 0.05, the child’s age, the mother’s IPARAN score, the family structure, the family’s net monthly income, and the mother’s nationality were included as potential confounders in the models for the child’s psychosocial health. 

In addition, the role of socio-demographic characteristics (e.g., family structure, the mother’s IPARAN score, family net monthly income, and the mother’s nationality) on the intervention’s effect was examined by adding an interaction term (e.g., [(study group) x (net monthly income)]) to the model. If the interaction term had a *p*-value < 0.10, a stratified analysis was performed. 

All statistical analyses were conducted using Statistical Package for the Social Sciences (SPSS) version 22 (IBM Corp. Released 2013. IBM SPSS Statistics for Windows, Version 22.0. Armonk, NY: IBM Corp.). 

## 3. Results

### 3.1. Sample Characteristics

In the entire study population, the mean IPARAN score for the mothers was 4.2 (SD = 2.9), the mean age of the mothers at baseline was 30.8 years (SD = 6.0 years), and 85% of the mothers were of Dutch nationality. The mean IPARAN score for the fathers was 2.3 (SD = 1.8), the mean age of the fathers at baseline was 33.1 years (SD = 6.5 years), and 86% of the fathers were of Dutch nationality. Twenty-one percent of the parents were single parents, and 31% of the parents had a net monthly income less than €1800. The mean age of the children at baseline was 6.2 months (SD = 3.3 months), and 43% of the children were girls. The demographics of the parents and children are summarized in Table 1.

Table 1 also specifies the general characteristics of the parents and children in the two study groups. The mothers in the intervention group had significantly higher IPARAN scores compared to the mothers in the control group (4.8 [SD = 3.4] vs. 3.7 [SD = 2.5], respectively; *t* = 3.05; *p* = 0.003); in contrast, the fathers in the intervention group had significantly lower IPARAN scores compared to the fathers in the control group (1.9 [SD = 1.9] vs. 2.5 [SD = 1.6]; *t* = −2.41; *p* = 0.017). In the intervention group, 73% of the mothers and fathers were of Dutch nationality, compared to 93% of the mothers (χ^2^ = 23.99; *p* < 0.001) and 95% of the fathers (χ^2^ = 27.64; *p* < 0.001) in the control group. In addition, the intervention group contained significantly more single parents compared to the control group (39% vs. 8%, respectively; χ^2^ = 42.29; *p* < 0.001), and the intervention group contained significantly more parents with low income, defined as net monthly income less than €1800 (55% vs. 16% in the control group; χ^2^ = 50.46; *p* < 0.001). Lastly, the children of the parents in the intervention group were older at baseline compared to the children of the parents in the control group (7.6 months [SD = 3.4 months] vs. 5.1 months [SD = 2.8 months], respectively; *t* = 6.95; *p* < 0.001).

### 3.2. Characteristics of Participants Lost-to-Follow Up

We found significant differences between parents who completed both questionnaires at baseline and follow up of 18 months and parents who did not respond to the 18-month follow-up questionnaire (i.e., lost-to-follow up). Specifically, compared to the parents who completed both questionnaires at baseline and 18 months time point, the parents lost-to-follow up contained higher percentages of non-Dutch mothers (χ^2^ = 9.76; *p* = 0.002), non-Dutch children (χ^2^ = 16.21; *p* < 0.001), single parents (χ^2^ = 16.83; *p* < 0.001), low educated parents (χ^2^ = 8.76; *p* = 0.013), and low net monthly income (χ^2^ = 17.96; *p* < 0.001). 

Disaggregated to study group, parents lost-to-follow up in the intervention group contained higher percentages of single parents (χ^2^ = 10.12; *p* < 0.001), non-Dutch children (χ^2^ = 7.53; *p* = 0.023), and low net monthly income (χ^2^ = 9.75; *p* = 0.002). Parents lost-to-follow up in the control group contained higher percentages of non-Dutch mothers (χ^2^ = 6.46; *p* = 0.011).

### 3.3. Effect of the Intervention on Empowerment

Table 2 summarizes the results with respect to parenting skills, social support (both from family and friends, and from the partner), self-sufficiency, and resilience at baseline and the 18-month follow-up for both study groups. Interestingly, neither group improved in any outcome from baseline to follow-up; at the 18-month time point, each score was either unchanged or was significantly (*p* < 0.05) worse compared to the respective baseline score. In both groups, the scores of competence as a parent and partner relationship were significantly worse at follow-up. 

Table 3 summarizes the effect of the Supportive Parenting Intervention on parenting skills, social support (both from family and friends, and from the partner), self-sufficiency, and resilience. The uncorrected model revealed a significant difference between the intervention and control groups for resilience. However, after adjusting the model for confounders, no significant difference remained between the two groups.

### 3.4. Effect of the Intervention on the Child’s Psychosocial Health and Behavior

Appendix A summarizes the effect of the Supportive Parenting Intervention on the child’s psychosocial health and behavior at the 18-month follow-up. After adjusting the model for confounders, we found no significant difference between intervention and control groups with respect to the CBCL total score (B = 0.72 [95% CI: −5.05; 6.50]; *p* = 0.806), internal score (B = 0.45 [95% CI: −1.59; 2.49]; *p* = 0.665), or external score (B = 0.00 [95% CI: −2.08; 2.09]; *p* = 0.997).

### 3.5. Evaluation of Interaction Effects by Child and Parental Characteristics

Exploratory interaction analyses were performed between the research groups and demographic characteristics. Significant interaction effects were found for income level, the mother’s IPARAN score, and family structure, and are summarized in Appendix A. After stratified analyses, only significant results (*p* < 0.05) were found for the research group and family structure. Single parents in the intervention group had lower competence utilization scores (i.e., these parents were less able to utilize their competencies) compared to single parents in the control group (B = −0.84 [95% CI: −1.58; −0.10]; *p* = 0.027).

### 3.6. Evaluation of the Supportive Parenting Intervention

In the follow-up questionnaire, the parents in the intervention group were asked their opinion regarding the intervention. Of the 85 parents who completed the follow-up questionnaire, 72 (85%) answered the questions regarding the evaluation. Table 4 summarizes the percentage of parents who answered either “agree” or “totally agree” in response to statements regarding what they learned from the Supportive Parenting Intervention. 

On average, the parents in the intervention group received five home visits. In open-ended questions, the parents were asked what they appreciated most about the intervention. The parents reported that they generally appreciated the fact that the visits occurred in the parents’ home, as this is considered a safe environment and the parents did not worry about disrupting their child’s sleeping or feeding schedule. In addition, the parents reported that they appreciated the fact that the Youth Health Care nurse provided the parents with their full undivided attention during the visits. Sixty-eight percent of the parents in the intervention group perceived the home visits as meaningful. Moreover, 67% of the parents in the intervention group reported that they felt more confident, and 65% reported that had more self-esteem after completing the Supportive Parenting Intervention. Finally, 52% of parents reported that they understood their child better after completing the intervention. 

## 4. Discussion

Here, we evaluated the effect of the Supportive Parenting Intervention with respect to empowering parents who are at risk for developing parenting stress. Specifically, we evaluated parenting skills, social support (both from family and friends, and from the partner), the parents’ self-sufficiency and resilience, and the child’s psychosocial health and behavior. In addition, we evaluated elements of the Supportive Parenting Intervention among parents in the intervention group. 

Overall, we found no effect of the intervention on either the parents’ empowerment or the child’s psychosocial health and behavior between baseline and follow-up. Our analysis revealed that neither the intervention group nor the control group improved with respect to any aspect of empowerment between baseline and follow-up. Similar results were obtained when we analyzed only first-time parents (data not shown). With respect to social contacts and partner support, and competence as a parent, the parents’ scores were either unchanged or were worse at follow-up (i.e., when the child was 18 months of age) compared to their baseline scores. 

We did observe decreased parental outcomes (i.e., Social Support, Partner Relation, Competence as a Parent) in both study groups between baseline and follow-up. These effects may be explained—at least in part—by the general effect of the child’s developmental phase on the parents’ sense of skills and confidence in terms of raising their child. In general, parenting changes as the child grows, and the toddler months (19–25 months of age) can be a particularly stressful period for parents [38]. During this period, the child enters a new developmental stage and this is often associated with new challenges for the parents. For example, toddlers often display more challenging behavior as they become better at understanding both their own emotions and the emotions of others. At this age, children learn to communicate their likes and dislikes, and they become capable of asserting their own will [39]. In their search for autonomy, they often test the limits set by their parents [38]. In a study by Fraser et al., the effect of a home-visit program to at-risk families shortly after childbirth was evaluated [40]. Consistently with our results, the authors found a reduction in the parents’ sense of competence 6 weeks after childbirth. However, by 12 months, the competence scores returned to the same values as measured at baseline [40]. Thus, the effect of the child’s developmental phases—and the corresponding changes in parental skills—warrant further investigation. Although parental confidence may diminish due to changes in the child’s developmental phase and the parental skills required during these phases, our intervention was unable to counteract the changes in parental confidence that occur as the child enters the toddler phase, as we found no differences between the intervention and control groups. 

Given the above-mentioned developmental changes that occur at this age, the duration of our study (until the child reached 18 months of age) may have been too short to detect relevant differences. Likewise, the Supportive Parenting Intervention itself can be extended to last beyond 18 months of age, thereby supporting parents beyond the toddler years and improving both their competences as a parent and their use of those competences.

Another potential explanation for not observing an effect of the intervention could be related to other family circumstances. For example, having a limited income. In the current study, half of parents in the intervention group had an income below €1800. For these parents, different types of support might be more appropriate. 

It should be noted that the parents in the control group were selected from an area in which the intervention had been performed until the start of this study (January 2014), but was no longer available. It is therefore possible that some of the YHC professionals in this area still used some elements of the intervention during the course of their work, which may have reduced any possible differences between the intervention and control groups. Moreover, the participants in the intervention and control group were not comparable at baseline with regard to social and demographic characteristics. The participants in the intervention group appeared to have a higher level of disadvantage, showing, for example, an average higher IPARAN score. Although we corrected for these differences in the analysis, they may have impacted the potential to observe effects. 

Although we found no positive effects of the intervention in terms of empowerment, the parents in the intervention group reported that they were generally satisfied with the intervention. In particular, they felt more confident in their role as a parent and had more self-esteem after the intervention. Moreover, the parents in the intervention group appreciated the personal attention received from the YHC nurse and the fact that the visits occurred in their own home. These results suggest that the Supportive Parenting Intervention likely has clear—albeit less tangible—benefits for the parents.

### Methodological Considerations 

In addition to the considerations discussed above, this study has methodological limitations that warrant discussion. Due to low response at the start of our study, we extended the inclusion period by 4 months, making the total inclusion period eight months. However, we were not able reach our target number of 196 participants in the intervention group; a total of 124 parents participated in the intervention group. We were able to include more participants in the control group than aimed for: 177 instead of 133. Taking into account lost-to-follow-up, the power of our study slightly decreased. Thereby, the socio-demographic characteristics of the two study groups were not distributed equally; specifically, the intervention group contained larger percentages of parents who were non-Dutch, had a low net monthly income, were single parents, and had higher IPARAN scores. These differences may have arisen due to the methods used to select participating YHC centers, which resulted in the participants in the control group living primarily in a suburban area and the participants in the intervention group living primarily in an urban area. In an attempt to account for this unequal distribution of characteristics, the relevant variables were added to the models as confounders. In addition, in some cases, the parents completed the baseline questionnaire after their first visit by a YHC nurse, which may have influenced their responses. Finally, it is possible that due to differences in practices of the nurses providing the intervention the effect of the intervention may have differed. However, the intervention is provided according to a set structure, which includes nurse training and an intervention manual describing the content of each visit. Therefore, variation in practices between nurses was expected to be low. Nevertheless, we recommend that future studies monitor the nurses’ practices in order to take into account potential variability.

Despite these limitations, a strength of this study lies in the fact that we were able to evaluate this intervention within a group of parents who were at risk for developing parenting stress, using a control group of parents who had the same risk but did not have access to the intervention. Nevertheless, this study should be repeated using a longer follow-up period, including the empowerment of parents as their child passes various developmental stages, thereby increasing the likelihood of detecting inherent changes in empowerment. 

## 5. Conclusions

We investigated the effect of the Supportive Parenting Intervention on the empowerment of parents who are at risk for developing parenting stress. Our results show that parents were generally satisfied with the intervention and had increased confidence in their parenting role after the intervention. Importantly, the parents appreciated the fact that a healthcare professional visited them at their home and provided the parents with their undivided attention. Although the parents were satisfied with the intervention and felt more confident in their role as a parent, this study shows no positive effect with respect to the indicators of parental empowerment. We recommend further research to strengthen the intervention and its application in daily practice and to evaluate it in a large randomized controlled trial. 

## Figures and Tables

**Figure 1 ijerph-16-04508-f001:**
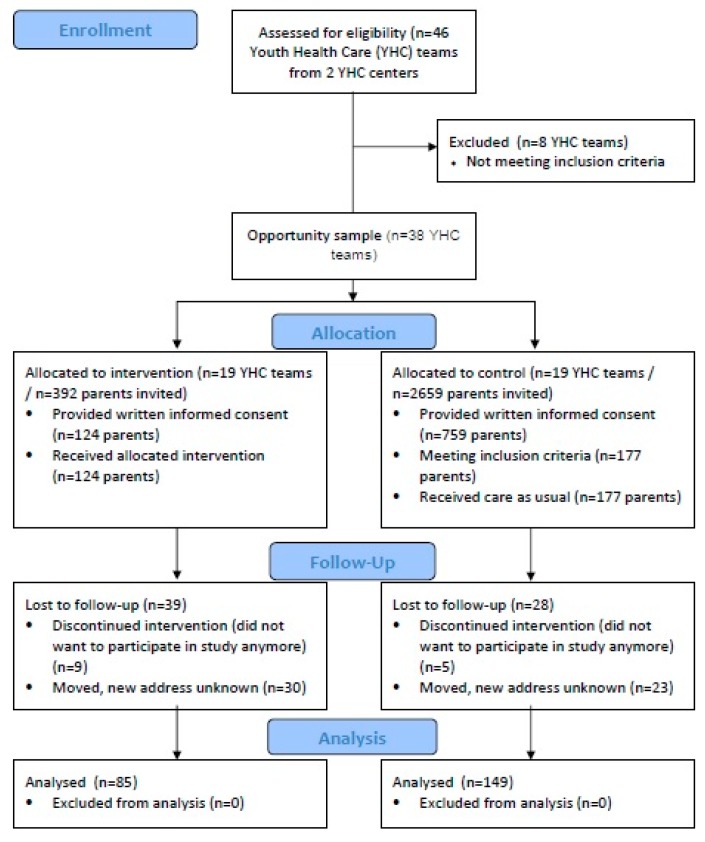
CONSORT flow chart describing the progress of participants through the trial.

**Table 1 ijerph-16-04508-t001:** Socio-demographic characteristics of the parents and their children at baseline.

Characteristic	Total (*n* = 301)	Intervention Group (*n* = 124)	Control Group (*n* = 177)	*p*-Value ^1^
Age of the child in months (mean [SD])	6.2 (3.3)	7.6 (3.4)	5.1 (2.8)	<0.001
Sex of the child (% girls)	43	47	41	0.293
Nationality of the child (% Dutch)	91	83	97	<0.001
Family structure (% single parents)	21	39	8	<0.001
Sex of the parent (% female)	92	90	93	0.667
Age of the mother in years (mean [SD])	30.8 (6.0)	31.0 (7.0)	30.7 (5.3)	0.623
Age of the father in years (mean [SD])	33.1 (6.5)	34.0 (7.3)	32.6 (6.0)	0.095
Nationality of the mother (% Dutch)	85	73	93	<0.001
Nationality of the father (% Dutch)	86	73	95	<0.001
Educational level of the mother (%)				0.603
Low	15	16	13	
Middle	49	50	48	
High	37	34	39	
Educational level of the father (%)				0.827
Low	14	16	13	
Middle	48	46	49	
High	38	38	38	
Net monthly family income (% <€1800)	31	55.4	15.5	<0.001
IPARAN score of the mother (mean [SD])	4.2 (2.9)	4.8 (3.4)	3.7 (2.5)	0.003
IPARAN score of the father(mean [SD])	2.3 (1.8)	1.9 (1.9)	2.5 (1.6)	0.017

Notes: IPARAN, Instrument for the identification of Parents At Risk for child Abuse and Neglect; SD, standard deviation. ^1^ Intervention group vs. Control group; *p*-values were calculated using a χ^2^-test or independent t-test.

**Table 2 ijerph-16-04508-t002:** Change in empowerment between baseline and the 18-month follow-up.

Outcome	Intervention Group				Control Group			
	*n*	Baseline Mean (SD)	Follow-Up Mean (SD)	*p*-Value ^2^	*n*	Baseline Mean (SD)	Follow-Up Mean (SD)	*p*-Value ^2^
Parenting skills								
Basic parenting skills ^1^	81	21.97 (3.72)	22.30 (3.05)	0.291	149	23.64 (2.97)	23.90 (2.61)	0.201
Social support								
Social contacts ^1^	84	14.68 (2.78)	14.80 (2.98)	0.658	149	16.38 (3.09)	16.08 (3.05)	0.081
Partner relationship ^1^	61	16.92 (3.05)	16.30 (3.20)	0.048	140	18.14 (2.50)	17.61 (2.63)	0.005
Self-sufficiency								
Competence as a parent ^1^	85	8.26 (1.37)	7.93 (1.31)	0.039	144	8.17 (1.28)	7.85 (1.17)	0.001
Resilience								
Resilience ^1^	82	78.67 (11.07)	80.65 (10.52)	0.063	146	82.62 (9.78)	82.16 (9.75)	0.450

Notes: SD, standard deviation; Bold numbers indicate a significant (*p* < 0.05) between baseline score and follow up score within study group. ^1^ High score indicates improved performance. ^2^ Baseline score versus Follow up score; *p*-values were calculated using a Paired Student’s *t*-test.

**Table 3 ijerph-16-04508-t003:** The association between research condition and primary outcome measures.

Outcome	Model 1 ^1^			Model 2 ^2^		
	Intervention Group vs. Control Group			Intervention Group vs. Control Group		
	*n*	Beta Coefficient (95% CI)	*p*-Value ^3^	*n*	Beta Coefficient (95% CI)	*p*-Value ^3^
Parenting skills						
Basic parenting skills	228	0.07 (−0.63; 0.77)	0.847	225	−0.50 (−1.15; 0.14)	0.126
Social support						
Social contacts	233	0.42 (−0.18; 1.02)	0.168	228	−0.08 (−0.69; 0.54)	0.810
Partner support	201	−1.00 (−0.77; 0.58)	0.779	198	−0.68 (−1.35; 0.00)	0.050
Self−sufficiency						
Competence as a parent	229	−0.01 (−0.34; 0.32)	0.933	225	0.10 (−0.21; 0.41)	0.519
Resilience						
Resilience	228	2.43 (0.25; 4.61)	0.029	223	1.32 (−0.90; 3.54)	0.244

Note: 95% CI, 95% Confidence Interval; old numbers indicate a significant (*p* < 0.05) difference between intervention and control group. ^1^ Model without correction for confounders. ^2^ Model corrected for baseline score, age of the child, nationality of the mother, and the mother’s IPARAN score. ^3^
*p*-values were calculated using a generalized linear model.

**Table 4 ijerph-16-04508-t004:** Parent appreciation of the intervention (*n* = 74).

“Because of the Supportive Parenting Intervention, I…”	% Totally Agree
Feel more confident as a parent	67
Have more self-esteem	65
Understand my child better	52
Am more easily inclined to ask for help	47
Learned to solve problems myself	43
Received more social support	40

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
