# Peer review of "A Controlled Trial in Community Pediatrics to Empower Parents Who Are at Risk for Parenting Stress: The Supportive Parenting Intervention"

_ijerph, 2019, doi:10.3390/ijerph16224508_

Round 1

Reviewer 1 Report

This paper deals with a subject of great interest such as the effectiveness of family support interventions carried out during the transition to parenthood. The authors have conducted a controlled trial with pre- and post-intervention in the setting of community pediatrics among parents. The work has an updated introduction of the topic and the design used is adequate. However, the results obtained do not allow us to confirm the effectiveness of the intervention carried out.

Although it is important to publish studies where the expectations of researchers are not confirmed, it is equally necessary to try to explain why the expected results are not obtained. In this case, it is important to try to explain why the analyzed program does not achieve the proposed objectives. 

In this sense, the authors should make an important effort to try to explain the results obtained. Specifically:

1) In addition to the methodological limitations pointed out by the authors themselves, they should explain why they have used a paired Student’s t-test instead of repeated-measures ANOVA with the interaction effect × Group, as is habitual to evaluate the effectiveness of the interventions. 

2) They should carry out a more conceptual analysis of the results that would allow them to obtain and include in the paper the practical implications of their study.

Author Response

Dear editorial office, dear reviewers,

We would like to thank the editor and the reviewers for their time and effort to read our manuscript and provide us with relevant and critical feedback to further improve this work.

We address each reviewers’ comment below and marked changes made to the manuscript using “track changes” function in word.

Kind regards, on behalf of all authors,

Amy van Grieken

REVIEWER I

We have revised the manuscript following the suggestions of the reviewer.

1) This paper deals with a subject of great interest such as the effectiveness of family support interventions carried out during the transition to parenthood. The authors have conducted a controlled trial with pre- and post-intervention in the setting of community pediatrics among parents. The work has an updated introduction of the topic and the design used is adequate. However, the results obtained do not allow us to confirm the effectiveness of the intervention carried out.

Although it is important to publish studies where the expectations of researchers are not confirmed, it is equally necessary to try to explain why the expected results are not obtained. In this case, it is important to try to explain why the analyzed program does not achieve the proposed objectives. 

In this sense, the authors should make an important effort to try to explain the results obtained. Specifically:

In addition to the methodological limitations pointed out by the authors themselves, they should explain why they have used a paired Student’s t-test instead of repeated-measures ANOVA with the interaction effect × Group, as is habitual to evaluate the effectiveness of the interventions. 

We thank the reviewer for acknowledging the importance of publishing these results. The paired t-test analysis presented in Table 2 is complemented with the Generalized Linear Models (GLM) analysis presented in Table 3. We presented the paired t-test in Table 2 to indicate the change between baseline and follow-up within both research groups. Model 1, presents the results from the GLM uncorrected for co-variates. Table 3, Model 2, presents the results corrected for co-variates.

We discussed with our statistician the use of ANCOVA (i.e. Model 2 in Table 3) versus the use of the Repeated Measures ANOVA. We believe that the use of ANCOVA is appropriate in this study as we have only one follow-up measure and this approach gains power (see for example McKenzie at http://siteresources.worldbank.org/DEC/Resources/Beyond_Baseline_and_FollowUpJDE_final.pdf, and the post by Ozler at http://blogs.worldbank.org/impactevaluations/why-difference-difference-estimation-still-so-popular-experimental-analysis).

2) They should carry out a more conceptual analysis of the results that would allow them to obtain and include in the paper the practical implications of their study.

Following the suggestions by the reviewer we elaborated in the discussion about the lack of effects observed.

(line 417-436): “Given the above-mentioned developmental changes that occur at this age, the duration of our study (until the child reached 18 months of age) may have been too short to detect relevant differences. Likewise, the Supportive Parenting Intervention itself can be extended to last beyond 18 months of age, thereby supporting parents beyond the toddler years and improving both their competences as a parent and their use of those competences.
Another potential explanation for not observing an effect of the intervention could be related to other family circumstances. For example, having a limited income. In the current study half of parents in the intervention group had an income below €1800. For these parents, different types of support might be more appropriate. 
It should be noted that the parents in the control group were selected from an area in which the intervention had been performed until the start of this study (January 2014), but was no longer available in that area. It is therefore possible that some of the YHC professionals in this area still used some elements of the intervention during the course of their work, which may have reduced any possible differences between the intervention and control groups. Also, the participants in the intervention and control group were not comparable at baseline with regard to social and demographic characteristics. The participants in the intervention group appeared to have a higher level of disadvantage, showing for example an average higher IPARAN score. Although we corrected for these differences in the analysis, they may have impacted the potential to observe effects. ”

Reviewer 2 Report

Thank you for the opportunity to review this manuscript and your work in community pediatrics to support parents. The manuscript is very well written, methodologically sound, and provides worthwhile contributions to public health. 

A few minor revisions provided below: 

Line 27: please spell out the CBCL acronym

Line 87-88: slightly confusing as written 

Line 114: after 329, change is to was 

Page 4, figure 1: the N on the control group side does not seem correct, based on the narrative I think it should be N=149. 

Line 193: there's an extra ) after resilience 

Tables 2 and 3: Resilience is listed twice.  I think the first time is the heading and the second time is the variable, but not clear based on the indention. 

Line 392: Please provide further explanation of this/ specify the parental outcome. 

Author Response

Dear editorial office, dear reviewers,

We would like to thank the editor and the reviewers for their time and effort to read our manuscript and provide us with relevant and critical feedback to further improve this work.

We address each reviewers’ comment below and marked changes made to the manuscript using “track changes” function in word.

Kind regards, on behalf of all authors,

Amy van Grieken

REVIEWER II

We revised the manuscript according to the suggestions provided by the reviewer.

  • Line 27: please spell out the CBCL acronym

We spelled out the CBCL acronym in the abstract.

  • Line 87-88: slightly confusing as written

We rewritten the sentences (line 85-89): “Parents who are at risk for developing parenting stress were compared to parents who had the same risk but received care as usual. We evaluated the intervention’s effects on parenting skills, social support (both from family and friends, and from the partner), self-sufficiency, and resilience. In addition, we examined the intervention’s effect on the child’s health, specifically child psychosocial health measured by the Child Behavior Check List (CBCL). “

  • Line 114: after 329, change is to was

We changes is to was in this sentence.

  • Page 4, figure 1: the N on the control group side does not seem correct, based on the narrative I think it should be N=149.

The n on the control group side should be 149, because of 177 participants at baseline and 28 lost-to-follow-up. The narrative is correct. However, we notified that in the figure the n provided in the box “ analysis” is incorrect for the control group. This should be 149. We changed this and upload an updated figure.

  • Line 193: there's an extra ) after resilience

We deleted the extra ).

  • Tables 2 and 3: Resilience is listed twice. I think the first time is the heading and the second time is the variable, but not clear based on the indention.

We adapted the first column to indicate heading and variable. We ask the editorial office to make the table in the appropriate format and judge the current use of indention.

  • Line 392: Please provide further explanation of this/ specify the parental outcome.

We specified the parental outcomes we were discussing here (line 392): “We did observe decreased parental outcomes (i.e. Social Support- Partner Relation, Competence as a Parent) in both study groups between baseline and follow-up.”

Reviewer 3 Report

It was a pleasure to review this manuscript.  There are many programs like the SPI offered around the world and very few are evaluated for anything other than satisfaction ratings.  Studies such as this one are important for improving understanding of the effectiveness of programs and also acknowledging the discrepancies between the satisfaction ratings of parents, who are often appreciative of any support offered, and objectively measured outcomes.

I had a few points for consideration - nothing major.

It would be helpful if the abstract referred to 'nurse' rather than 'professional' as it gives a clearer indication of the type of professional involved.

Line 28 Abstract: Change it's to its.  One of the strange exceptions in English because it is a possessive but it doesn't have an apostrophe.

Line 36: Could experiences be changed to outcomes or another similar word to avoid repetition of the word experience in this sentence?

I couldn't find details of the randomization of teams, but it is included in the CONSORT flowchart.  A paragraph describing the procedure would be helpful.

The content of lines 146-147 and lines 66-68 looks the same or very similar.  One of these should be removed.

On page 11, lines 410-412, a point was made regarding the duration potentially being too short. If possible, it might be useful to incorporate this point into the last sentence of the abstract.  It's an important point and could potentially be lost.

Page 10, line 375.  Change 'that' to 'they'.

There is some repetition with Table 4 and the text directly above the Table.  I recommend removing/modifying the text directly above the table to avoid repetition. 

Who administered the questionnaires?  Was it the nurse visiting the family?  If so, could that have influenced the responses, especially to items mentioned in 2.6.3?

Figure 1 should have an informative label and doesn't need to have the CONSORT branding at the top.  See https://www.mdpi.com/1660-4601/16/20/3810/htm for an example from an article published in IJERPH.

Author Response

Dear editorial office, dear reviewers,

We would like to thank the editor and the reviewers for their time and effort to read our manuscript and provide us with relevant and critical feedback to further improve this work.

We address each reviewers’ comment below and marked changes made to the manuscript using “track changes” function in word.

Kind regards, on behalf of all authors,

Amy van Grieken

REVIEWER III

We have revised the manuscript following the suggestions provided by the reviewer.

  • It would be helpful if the abstract referred to 'nurse' rather than 'professional' as it gives a clearer indication of the type of professional involved.
  • Line 28 Abstract: Change it's to its.  One of the strange exceptions in English because it is a
  • possessive but it doesn't have an apostrophe.
  • Line 36: Could experiences be changed to outcomes or another similar word to avoid repetition of the word experience in this sentence?

We followed the reviewer’ suggestion and changed professional to nurse in the abstract, changed it’s  to its, and experiences to outcomes.

  • I couldn't find details of the randomization of teams, but it is included in the CONSORT flowchart.  A paragraph describing the procedure would be helpful.

We thank the reviewer for noticing that “randomized’ is depicted in the flow chart. As we describe in the manuscript text (line 101-110), this was an opportunity sample. The allocation of the teams too either intervention or control group was based on the region the Youth Health Care was active in. No randomization took place. We adapted the flow chart to match the procedure. 

  • The content of lines 146-147 and lines 66-68 looks the same or very similar.  One of these should be removed.

We removed this sentence from the introduction paragraph (line 66-68).

  • On page 11, lines 410-412, a point was made regarding the duration potentially being too short. If possible, it might be useful to incorporate this point into the last sentence of the abstract.  It's an important point and could potentially be lost.

We thank the reviewer for this suggestion. We incorporated the point on the duration of the intervention into the last sentence of the abstract (line 28-30): “We recommend research to strengthen the intervention and its application in daily practice, for example by increasing the intervention duration, and to evaluate it in a large randomized controlled trial.”

  • Page 10, line 375.  Change 'that' to 'they'.
  • There is some repetition with Table 4 and the text directly above the Table.  I recommend removing/modifying the text directly above the table to avoid repetition. 

We changed the text according to these suggestions.

  • Who administered the questionnaires?  Was it the nurse visiting the family?  If so, could that have influenced the responses, especially to items mentioned in 2.6.3?

To clarify that the nurse was not present at the time the parents completed the questionnaire we added this information to 2.2. data collection “Parents completed the baseline and follow-up questionnaire in their own time, the nurse was not present.” However, as we mentioned in the discussion section methodological considerations, some parents completed the questionnaire after the visit of the nurse. This could have impacted the responses provided.

  • Figure 1 should have an informative label and doesn't need to have the CONSORT branding at the top.  See https://www.mdpi.com/1660-4601/16/20/3810/htm for an example from an article published in IJERPH.

We thank the reviewer for this suggestion. We provide an informative label and upload an updated version of the figure without the CONSORT branding at the top.